# Transcript PHF19-207 as a Potential Biomarker for Colon Cancer Diagnosis and Screening

**DOI:** 10.3390/biom15060766

**Published:** 2025-05-26

**Authors:** Stefan Kmezic, Sandra Dragicevic, Tamara Babic, Jelena Ljubicic, Ivan Dimitrijevic, Aleksandra Nikolic, Velimir Markovic

**Affiliations:** 1Clinic for Digestive Surgery, University Clinical Center of Serbia, dr Koste Todorovica 6, 11 000 Belgrade, Serbia; ivanclean@gmail.com (I.D.); mbecmbeca@yahoo.com (V.M.); 2Faculty of Medicine, University of Belgrade, dr Subotica 8, 11 000 Belgrade, Serbia; 3Gene Regulation in Cancer Group, Institute of Molecular Genetics and Genetic Engineering, University of Belgrade, Vojvode Stepe 444a, 11 042 Belgrade, Serbia; sandra.dragicevic@imgge.bg.ac.rs (S.D.); tamara.babic@imgge.bg.ac.rs (T.B.); aleksandra.nikolic@imgge.bg.ac.rs (A.N.); 4Clinic for Allergy and Immunology, University Clinical Center of Serbia, dr Koste Todorovica 2, 11 000 Belgrade, Serbia; jelena.ljubicic.mfub@gmail.com

**Keywords:** cancer diagnosis, cancer screening, colon cancer, gene expression, PHF19, transcript

## Abstract

A recent comprehensive pan-cancer study indicated the high translational potential of the transcript PHF19-207 as a biomarker for colon cancer. This study aimed to analyze the expression of PHF19-207 in colon tissue samples from two different settings to evaluate its clinical utility for diagnosis and screening. Surgical samples of colon tumor and non-tumor tissue were analyzed to determine the diagnostic value of PHF19-207 and its potential correlation with tumor characteristics. Additionally, biopsied samples from individuals undergoing national colorectal cancer screening were examined to assess the potential use of PHF19-207 in early detection. PHF19-207 expression levels were measured in all samples using Real-Time Polymerase Chain Reaction. A statistically significant difference was observed between tumor and non-tumor tissue (*p* = 0.002) and between tumor tissue and healthy mucosa samples (*p* < 0.001). Furthermore, polyp samples exhibited significantly higher PHF19-207 expression compared to healthy mucosa (*p* = 0.035). Receiver operating characteristic (ROC) analysis indicated that PHF19-207 can effectively differentiate malignant from healthy tissue, with an AUC value of 0.9044. Considering the increasing incidence of colorectal cancer in younger populations and the need for improved early detection, PHF19-207 expression could be explored as the basis for a relatively simple and efficient test, enabling a more comprehensive and affordable screening strategy.

## 1. Introduction

Colorectal cancer (CRC) is a major global health concern and ranks as the third most common cancer and the second leading cause of cancer-related deaths across both sexes, according to GLOBOCAN (Global Cancer Observatory) data. Approximately 70% of colorectal carcinomas are diagnosed in the colon, which remains the primary site of disease development [1]. Despite advancements in prevention and early detection, CRC continues to present a significant global burden, particularly in developing countries, where access to screening and early intervention may be limited [2,3]. The implementation of population-based screening programs has proven effective in reducing both the incidence and mortality associated with CRC [3]. These programs are currently recommended for individuals between the ages of 50 and 75 years. However, in response to emerging evidence of an increasing incidence of CRC in younger populations, the American Cancer Society has recently suggested lowering the recommended starting age for screening to 45 years [4]. This shift highlights the ongoing need to refine strategies for early detection and diagnosis. The decline in CRC mortality among older populations has been attributed not only to screening programs but also to significant progress and innovations in the multidisciplinary treatment of colon cancer. However, epidemiological trends suggest that by the end of 2030, approximately 11% of colon cancer cases will be diagnosed in individuals younger than 50 years old, underscoring the importance of adapting screening and treatment strategies accordingly [5].

The early detection of CRC is critical for improving survival rates, as symptoms are often absent or nonspecific during the early stages of the disease. Symptoms tend to develop only as the cancer progresses, making it difficult to identify the disease before it reaches more advanced stages [6]. The management of CRC is highly dependent on preoperative diagnostic evaluation and the Tumor–Node–Metastasis (TNM) staging system [7]. The survival rate of patients with low-stage cancer is higher compared to high-stage cancers. The gold standard for colon cancer staging is computed tomography (CT), and virtual CT colonoscopy can also detect cancer lesions as well as perform local staging. Computed tomography colonography (CTC) has an accuracy of around 78–92% [8]. In this context, there is a growing focus on identifying novel molecular biomarkers that could facilitate the early detection of CRC, even before the onset of clinical symptoms [9].

Biomarkers represent measurable biological molecules in the body, such as protein, DNA, or RNA, that can be used to detect or confirm specific disease conditions [9,10]. Diagnostic biomarkers are typically used to confirm cancer in patients with visible signs of disease, while screening biomarkers are employed for disease detection in asymptomatic individuals, often as part of routine health check-ups or mass screening efforts. Both types of biomarkers must exhibit high specificity and sensitivity to provide reliable results and are usually employed in combination with other diagnostic procedures to confirm a final diagnosis.

Currently, several tumor markers, including carcinoembryonic antigen (CEA) and carbohydrate antigen 19-9 (CA 19-9), are utilized for monitoring disease progression and recurrence in CRC patients [3]. However, these markers have limitations, primarily due to their lack of sufficient specificity and sensitivity, which restricts their utility as diagnostic tools for early cancer detection. As a result, there is a continued need for the development of more effective biomarkers that can be used for earlier and more accurate diagnosis. Nowadays, there is a need to identify novel tumor markers such as Cytokeratin fragment 19 antigen (CYFRA 21-1), which has shown high sensitivity in the initial staging of colon cancer [11].

Recent research has highlighted the potential of transcriptional alterations as early events in the malignant transformation process [12]. Dysregulated gene expression resulting from the activation of alternative gene promoters has gained attention as a promising avenue for biomarker discovery. Alternative promoters can lead to the production of distinct transcript isoforms, affecting gene regulation, protein function, and disease progression [13,14]. Among the genes implicated in colon cancer, a comprehensive analysis of available transcriptomic data indicated the *PHF19* gene as being a potential candidate [14]. The *PHF19* gene encodes the PHD finger protein 19 (PHF19), which has been shown to play a tumor-promoting role in several cancer types, including CRC [15]. Notably, this gene has two alternative promoters that give rise to transcript variants, which are differentially expressed in malignant versus non-malignant colon tissue samples [14]. The *PHF19* gene promoter found to be upregulated in colon cancer tissue gives rise to the transcript PHF19-207 (Transcript ID: ENST00000456291). According to the Ensembl database, the PHF19-207 transcript is annotated as a protein-coding isoform with an incomplete coding sequence (CDS) (the PHF19-207 transcript structure is provided in Appendix A). Previous in silico analysis has indicated a low coding potential for the PHF19-207 transcript, suggesting that it may function primarily as a non-coding or partially coding RNA species with putative regulatory roles [16]. Despite its protein-coding annotation, the PHF19-207 transcript is translated into a truncated polypeptide lacking the critical domains present in the full-length canonical PHF19 protein, thereby raising questions about its capacity to exert full protein function. This raises the possibility that the PHF19-207 transcript may serve a regulatory function, potentially acting through RNA-mediated mechanisms such as competitive endogenous RNA interactions, the modulation of RNA-binding protein activity, or the regulation of transcriptional and post-transcriptional processes [17]. Furthermore, increased expression of the PHF19-207 transcript was observed in malignant colon cancer cell lines compared to a non-malignant control, further supporting its potential role in colon cancer pathophysiology [16].

Importantly, an emerging area of cancer research focuses on the role of non-coding RNA in tumorigenesis. While much attention has traditionally been placed on protein-coding genes, a growing body of evidence underscores the significant regulatory roles played by non-coding transcripts in cancer biology. Non-coding RNA molecules, such as long non-coding RNAs (lncRNAs) and microRNAs (miRNAs), have been implicated in a variety of cellular processes, including cell proliferation, apoptosis, metastasis, and immune evasion—all critical aspects of cancer development [18]. These non-coding transcripts often arise from protein-coding genes and can exert influence through mechanisms such as modulating gene expression, interacting with chromatin, and influencing RNA processing [18]. In many cases, these non-coding RNAs are dysregulated in cancer, making them promising candidates for use as biomarkers for early diagnosis and therapeutic targeting [18,19].

The PHF19-207 transcript (888 bp in length), as an lncRNA, may serve as an example of this emerging trend. Although the *PHF19* gene is known to encode a protein with tumor-promoting properties, the specific role of its non-coding transcript PHF19-207 is unknown. Preliminary evidence suggests that the PHF19-207 transcript is upregulated in colon cancer tissue, raising the possibility that this non-coding transcript may contribute to the molecular pathogenesis of colon cancer by modulating the key regulatory networks involved in tumor development [14,17]. Given its differential expression in malignant versus normal tissue, the PHF19-207 transcript may serve as a valuable screening biomarker for early-stage colon cancer.

In the current study, we focused on evaluating the clinical relevance of the PHF19-207 transcript in patient-derived colon tissue samples. The aim of this study was to analyze the expression of the PHF19-207 transcript in various types of colon tissue samples and to assess its potential as a biomarker for the early diagnosis of colon cancer. By investigating the role of non-coding RNA in the context of colon cancer, this research sought to further our understanding of the molecular underpinnings of CRC and contribute to the development of more accurate and accessible diagnostic tools.

## 2. Materials and Methods

### 2.1. Subjects

This study included two groups of participants recruited at the Clinic for Digestive Surgery-First Surgical Clinic, University Clinical Center of Serbia. Ethical approval was obtained from the Ethics Committee of the Faculty of Medicine—University of Belgrade, in accordance with the Declaration of Helsinki (25/XI-1; 20 November 2024). Informed consent was obtained from all participants.

The first study group consisted of 30 patients (age 54–83 years, 73% males) who underwent surgery for colon cancer between April 2023 and February 2024. The inclusion criteria for patients were as follows: histologically confirmed colon adenocarcinoma, absence of distant metastases, no preoperative treatment, and having signed informed consent. Patients diagnosed with rectal cancer or those who declined to provide informed consent were excluded from this study. The second group included 23 asymptomatic individuals (age 37–80 years, 56% males) enrolled through the national CRC screening program conducted on an annual basis, with recruitment occurring between October 2022 and October 2023. These individuals were referred for colonoscopy following a positive fecal occult blood test.

Paired colon tumor and surrounding non-tumor tissue samples were collected from all patients during the surgical removal of the tumor. Adenocarcinoma was further confirmed histopathologically, and tumor staging was determined according to the American Joint Committee on Cancer (AJCC) TNM staging criteria.

Samples of colon mucosa were collected from individuals from the national CRC screening program during the colonoscopy procedure in the outpatient setting. Based on the histopathological analysis, the samples were classified as polyposis tissue, healthy inflamed mucosa, or healthy non-inflamed mucosa.

All tissue specimens were immersed straightaway into RNAlater^®^ RNA Stabilization Solution (Thermo Fisher Scientific, Waltham, MA, USA) and stored at −80 °C until RNA isolation.

### 2.2. Relative Quantification of the PHF19-207 Transcript Expression

Total RNA was isolated from all tissue samples by PureLinkTM RNA Mini Kit (Thermo Fisher Scientific, Waltham, MA, USA) according to the manufacturer’s protocol. The concentration and purity of isolated RNA were determined by absorption at 260 and 280 nm using a BioSpec-nano spectrophotometer (Shimadzu Corporation, Kyoto, Japan).

The reverse transcription of total RNA (2 μg) was performed using the High-capacity cDNA Reverse Transcription Kit (Applied Biosystems, Foster City, CA, USA) following the manufacturer’s instructions. The reaction conditions were 10 min at 25 °C, 120 min at 37 °C, and 5 min at 85 °C.

The relative expression of the PHF19-207 transcript was measured by Real-Time Polymerase Chain Reaction (RT-PCR) using Power SYBR Green PCR Master Mix (Thermo Fisher Scientific, Waltham, MA, USA). For all measurements the expression of the glyceraldehyde-3-phosphate dehydrogenase (*GAPDH*) gene was used as an endogenous control. Melting curve analysis was performed to confirm the specificity of the obtained products. The sequences of used primers were as follows: PHF19-207 forward 5′-GATAGTCACAACACCAGGTGCC′3 and reverse 5′-CTTCCCCTGACACTGGCTCC-′3, and *GAPDH* forward 5′-GTGAAGGTCGGAGTCAACG-‘and reverse 5′-TGAGGTCAATGAAGGGGTC-′3.

The relative quantification was performed on a 7500 Real-Time PCR instrument (Applied Biosystems, Foster City, CA, USA) at the following reaction conditions: 2 min at 50 °C and 10 min at 95 °C, followed by 40 cycles of 15 s at 95 °C, and 1 min at 60 °C. All measurements were performed in triplicate and relative quantification was calculated with the 2^−dCt^ method where dCt  =  Ct_target gene_ − Ct_housekeeping gene_.

### 2.3. Analysis of Publicly Available Sequencing Data

To evaluate the expression of the PHF19-207 transcript in malignant colon tissues compared to non-malignant tissues, we used the UCSC Xena Browser platform (https://xenabrowser.net; accessed on 14 March 2025). The analysis was based on RNA sequencing (RNA-seq) data from The Cancer Genome Atlas (TCGA) and the Genotype-Tissue Expression (GTEx) databases. Transcript expression levels were quantified as log2(FPKM + 0.001) using the RSEM method. FPKM (Fragments Per Kilobase of transcript per Million mapped reads) is a normalization metric that accounts for sequencing depth and gene length, making expression levels comparable across samples. RSEM (RNA-Seq by Expectation-Maximization) is a computational method used to estimate gene and transcript abundances from RNA-seq data, improving quantification accuracy.

### 2.4. Statistical Analysis

GraphPad Prism v. 10.1.1. (GraphPad Software, LLC, Boston, MA, USA) was used for statistical analysis. Data were presented as percentages and mean ± SD (standard deviation). Statistical tests were selected based on the distribution of continuous data, which was assessed using the Shapiro–Wilk test. The differences between the data obtained for matched samples were tested by the Wilcoxon matched pairs signed rank test, while for independent samples, the Mann–Whitney U test and Kruskal–Wallis test were used. Receiver operating curve (ROC) analysis and area under curve (AUC) were used to analyze the degree of discrimination between two variables. A non-parametric Spearman’s rank correlation coefficient (r_s_) was applied to assess the correlation between variables. *p* values less than 0.05 were considered statistically significant.

## 3. Results

### 3.1. Patient Characteristics

The expression levels of the PHF19-207 transcript were analyzed in tissue samples of the gut mucosa obtained from patients with colon cancer and healthy individuals. The patient group consisted of 30 individuals who underwent surgery for colon cancer (mean age: 67.7 ± 7.2 years; 73% male). Their demographic and clinical data are presented in Table 1. The group of healthy individuals who underwent colonoscopy within the national screening program for CRC consisted of 23 individuals (mean age: 66.0 ± 10.0 years; 54% male). Notably, four cases in the screening group were diagnosed with intestinal polyps.

As shown in Table 1, the patient cohort represents an older population with a predominance of males and advanced-stage colon cancer. Tumors were more frequently located in the right colon, and a majority were classified as T3 or T4, indicating a focus on more severe cases. Lymph node involvement and lymphatic invasion were present in a substantial proportion of patients, with a high prevalence of perineural invasion (93.3%), suggesting an aggressive tumor profile. Venous invasion was observed in 43.3% of cases, further indicating a significant risk of metastasis.

### 3.2. The Expression of the PHF19-207 Transcript in Colon Cancer and Healthy Tissue Samples

A statistically significant difference was observed in the expression levels of the PHF19-207 transcript between tumor and non-tumor tissue samples (*p* = 0.002, Figure 1A). In the majority of patients (87%), the expression levels were higher in tumor tissue compared to paired non-tumor tissue samples, with a variable change ranging from 0.3- to 7.7-fold. In contrast, in four cases, a decrease in expression was observed (between 1.5 -and 3.4-fold change). When comparing samples of tumor tissue to samples of healthy intestinal mucosa, the differential expression of the PHF19-207 transcript was even more pronounced (*p* < 0.001, Figure 1B). Additionally, a significantly higher expression of the PHF19-207 transcript was found in polyp samples compared to samples of healthy intestinal mucosa (*p* = 0.035, Figure 1B).

There was no statistically significant difference between the relative abundance of the PHF19-207 transcript in healthy intestinal mucosa depending on the presence of inflammation (*p* = 0.472), but there was a slight increase in values observed in inflamed mucosa (Figure 1B).

No association was observed between the PHF19-207 transcript expression and tumor T or N stage ((T2: 2^−dCt^ = 0.0014 ± 0.0021, T3: 2^−dCt^ = 0.0003 ± 0.0005, T4: 2^−dCt^ = 0.0001± 0.0001; *p* = 0.237) and (N0: 2^−dCt^ = 0.0005 ± 0.0012, N1: 2^−dCt^ = 0.0004 ± 0.0005, N2: 2^−dCt^ = 0.0001 ± 0.0001; *p* = 0.497)). No significant correlation was found between serum CEA levels and the PHF19-207 transcript expression in tumor tissues (r_s_ = −0.228, 95% CI = −0.5519–0.1544, *p* = 0.224).

### 3.3. Validation of the PHF19-207 Transcript Expression Using Publicly Available Data

To further investigate the PHF19-207 transcript expression, we performed an independent analysis using the UCSC Xena Browser platform. The analysis revealed a significantly higher expression of the PHF19-207 transcript in primary tumor tissues compared to solid normal tissues from colon cancer patients (TCGA: primary tumor vs. solid tissue normal, *p* = 3.511 × 10^−12^). Additionally, when comparing the PHF19-207 transcript expression in primary tumor tissues from colon cancer patients to normal intestinal mucosa from healthy individuals, a markedly increased expression was observed in the tumor tissues (TCGA: primary tumor vs. GTEx: normal tissue, *p* = 1.442 × 10^−26^).

### 3.4. Diagnostic Potential of the PHF19-207 Transcript in Colon Cancer

To assess the significance of the PHF19-207 transcript as a diagnostic biomarker for colon cancer, we performed ROC analysis using expression data from tumor tissue samples obtained during surgery and healthy intestinal mucosa biopsies from individuals in the CRC screening program (Figure 2). The obtained results indicated the good performance of the PHF19-207 transcript expression for the discrimination of tumor from non-tumor tissue (AUC = 0.7344, 95% CI = 0.5983–0.8706, *p* < 0.0018). Notably, the diagnostic performance was even higher when distinguishing tumor tissue from healthy intestinal mucosa, with an AUC of 0.9044 (95% CI = 0.8179–0.9908, *p* < 0.0001).

## 4. Discussion

This study was designed as a targeted, exploratory analysis focusing on expression patterns in two groups: colon cancer patients and individuals from the national CRC screening program. The primary objective was to provide an initial assessment of the clinical applicability of the PHF19-207 transcript, building on existing evidence that it is significantly overexpressed in malignant colon tissue compared to non-malignant tissue. To explore its clinical relevance, the abundance of this transcript was analyzed in samples from two different settings. Surgical samples of tumor and adjacent non-tumor tissue from non-metastatic, treatment-naive colon cancer patients were analyzed to evaluate the diagnostic potential of the PHF19-207 transcript and its association with tumor characteristics. Patients with rectal cancer were intentionally excluded to minimize clinical heterogeneity and ensure a more uniform cohort in terms of tumor localization. Biopsied samples from individuals undergoing national screening for CRC were used to investigate the potential of the PHF19-207 transcript for screening purposes [20]. There was no significant difference in the average age of the analyzed groups, but they differed significantly according to the gender composition. There were more women among the individuals from the screening group than in the surgical patient group (44% vs. 27%). This difference might suggest a gender-related variation in screening participation or in the early biological factors influencing disease onset.

The relative abundance of the PHF19-207 transcript was initially evaluated in matched tumor and adjacent mucosa samples collected during surgery. The inclusion of treatment-naive patients ensured that expression levels reflected intrinsic tumor biology, unaffected by prior therapy. A statistically significant difference in the PHF19-207 transcript expression levels was observed between tumor and non-tumor tissue samples, with the majority of patients exhibiting elevated expression in tumor tissue. This observation was further supported by an independent analysis using publicly available data from the UCSC Xena Browser platform. The Xena dataset confirmed a significantly higher expression of the PHF19-207 transcript in primary tumor tissue compared to adjacent normal tissue samples. These results indicate the strong potential of the PHF19-207 transcript as a diagnostic tool. Although preliminary in vitro data indicated higher expression levels in cell lines derived from advanced-stage colon tumors, our study did not find a significant association between the PHF19-207 transcript expression and the T or N stage in patient samples [16]. This finding may result from the contribution of stromal cells to the tumor tissue transcriptome [21,22]. Given that the tumor–stroma ratio in colon tumors can be highly variable, the contribution of the stromal cells to the expression of any specific transcript in the tumor tissue can also vary considerably [23,24].

When comparing the PHF19-207 transcript expression levels in biopsied mucosa samples from individuals undergoing national CRC screening to surgical tumor tissue samples, we observed a significant difference. As expected, the expression of the PHF19-207 transcript in biopsied mucosa samples was lower compared to that in tumor tissue samples, which was aligned with the results from Xena Browser platform. Additionally, a lower expression of the PHF19-207 transcript in samples of healthy mucosa was observed in comparison to non-tumor tissue samples from colon cancer patients, highlighting the complexity of its potential application in early-stage detection. This observed difference could be due to transcriptomic and proteomic alterations in the tissue adjacent to the tumor, which differs from healthy mucosa [25,26]. Differences in the transcriptome of non-inflamed and inflamed gut mucosa were also observed in some pathological contexts [27]. In this study, we found slightly increased the PHF19-207 transcript expression levels in inflamed vs. non-inflamed gut mucosa, suggesting the involvement of the stromal component of the tumor in the obtained expression profiles. These findings also support the potential use of the PHF19-207 transcript as a diagnostic tool, but at the same time indicate that it cannot be used as a single biomarker due to potential interference with systemic or local inflammation.

The main limitation of this study was its relatively small sample size, which resulted from the strict inclusion criteria. This study was specifically designed to evaluate a pre-identified biomarker as a potential screening tool, and therefore included only treatment-naive colon cancer patients without metastatic disease, as they are representative of cases that would typically be identified through screening programs. This pilot study was conducted at a single department within a tertiary surgical center, where a large proportion of CRC cases involve either locally advanced rectal cancer treated with neoadjuvant therapy or metastatic disease, thus limiting the pool of eligible participants. Nevertheless, the consistent and statistically significant elevation of the PHF19-207 transcript in tumor tissue compared to matched non-tumor or healthy mucosa samples suggested strong discriminatory potential. The obtained AUC value of 0.9044 supported the potential of the PHF19-207 transcript as a biomarker candidate and indicated that further validation may be beneficial [28]. The lack of a statistically significant correlation between serum CEA levels and the PHF19-207 transcript expression in tumor tissues suggested that the PHF19-207 transcript may function as a biomarker independent of CEA. However, this result should be re-evaluated in a larger cohort due to the observed weak inverse relationship. Furthermore, the overlapping expression levels seen in inflamed mucosa and premalignant polyps underscore a key limitation in distinguishing these conditions. This suggests that the PHF19-207 transcript alone may not be sufficient for early-stage detection and should ideally be evaluated in conjunction with other clinical or molecular markers. In our study, such results were not available for consideration, since the protocol in the outpatient clinic that managed individuals from the screening did not include any analyses other than colonoscopy.

While this study highlighted the diagnostic potential of the PHF19-207 transcript, differentiating between malignant and non-malignant tissues, it also emphasized the need for caution when interpreting its levels in contexts involving inflammation or non-malignant changes, such as polyps. The slight increase in the PHF19-207 transcript expression in the inflamed gut mucosa should be further investigated before translating this biomarker into clinical practice. Since it appears that inflammation affects the usability of the PHF19-207 transcript as a single marker, combined analysis with acute phase inflammation parameters, such as C-reactive protein, erythrocyte sedimentation rate, and fibrinogen level, could be considered and explored as an alternative. Given the shift toward younger demographics for CRC screening, this strategy could offer a relatively simple and efficient test as part of a more comprehensive and affordable screening program.

This study suggests that the PHF19-207 transcript may serve as a marker of colon malignancy, regardless of whether it has a direct functional role. According to previous data, the increased expression of the PHF19-207 transcript in tumors can be attributed to tumor cells, but the exact cellular origin of the observed differences remains unclear [16]. Given its potential role as an lncRNA, the cellular and subcellular localization of the PHF19-207 transcript could indicate a possible regulatory function. Since PHF19 contributes to chromatin remodeling through its PHD finger domain, the PHF19-207 transcript, as non-coding variant, may similarly function in the nucleus by regulating gene expression via interactions with chromatin or RNA-binding proteins. Conversely, if located in the cytoplasm, it might function through post-transcriptional mechanisms, such as modulating RNA stability or acting as a miRNA sponge to regulate oncogene expression [17,18]. The PHF19-207 transcript may be involved in tumor initiation or maintenance by influencing the key signaling pathways associated with cell proliferation, apoptosis resistance, or immune evasion. These diverse potential roles emphasize the need for further research to identify its downstream targets and clarify the underlying molecular mechanisms. Importantly, future research should also investigate whether the PHF19-207 transcript expression is associated with tumor progression, which could provide deeper insight into its biological significance and reveal its potential as a prognostic biomarker or therapeutic target in colon cancer. Understanding whether the PHF19-207 transcript exerts a regulatory function in cancer-related cellular processes could shed light on its biological role and reveal whether its modulation may have therapeutic potential.

Moreover, the PHF19-207 transcript might also be suggested to participate in modulating the inflammatory response in the tumor microenvironment. Its elevated expression in inflamed compared to non-inflamed mucosa suggests that local inflammation may influence its expression. The link between inflammation and cancer is well-established, with chronic inflammation often driving tumorigenesis and influencing tumor progression [12,29]. The PHF19-207 transcript may be involved in this process by regulating immune cell recruitment, cytokine production, or other inflammatory mediators. If confirmed, this link would imply clinical relevance for the PHF19-207 transcript as a marker related to inflammation-associated premalignant lesions and as a potential target for modulating inflammation to reduce tumor progression. The interaction between the tumor and stroma, along with the contribution of immune cells to the cancer microenvironment, could further influence its expression, making the PHF19-207 transcript a complex marker shaped not only by tumor cells but also by the surrounding tissue environment. This concept requires further investigation, particularly to clarify the extent to which the PHF19-207 transcript expression is influenced by local inflammation or other non-tumor cells present in the colon tissue. Taken together, these observations support the potential of the PHF19-207 transcript as a biomarker candidate, while its clinical utility as a therapeutic target remains to be established.

Beyond its potential as a diagnostic marker, the PHF19-207 transcript may offer notable advantages, particularly regarding its non-invasive nature. Traditional biopsy techniques, which involve obtaining tissue samples directly from tumors, can be invasive and may carry associated risks. Furthermore, such procedures may not always be feasible, especially for tumors that are difficult to access or located in deep tissues. In contrast, liquid biopsies that detect circulating tumor RNA offer a less invasive means of obtaining molecular data, potentially eliminating the need for surgical procedures [30]. Detecting RNA molecules such as the PHF19-207 transcript in circulation may enable earlier disease recognition, potentially before symptoms emerge or imaging reveals pathological changes. Moreover, regular monitoring of circulating RNA levels might allow for tracking disease dynamics and adjusting therapeutic strategies accordingly. However, broader validation studies are required to establish the clinical utility of RNA biomarkers in the bloodstream, including the PHF19-207 transcript. As tumors tend to be heterogeneous, RNA detection in the peripheral circulation might offer a more comprehensive view of tumor biology, potentially capturing genetic or molecular profiles that might be missed with a single tissue biopsy. This broader understanding could pave the way for more targeted and effective treatment strategies.

The potential of the PHF19-207 transcript as a biomarker for colon cancer screening deserves further investigation, especially with regard to its applicability in non-invasive testing methods. Although RNA tends to be less stable than DNA or proteins in the bloodstream, it has been shown to remain detectable for up to 24 h following sampling [31,32]. Since the PHF19-207 transcript is present at low levels in healthy cells but is significantly elevated in tumor tissues, its presence in cell-free RNA may serve as an indicator of malignancy. This opens the possibility for using the PHF19-207 transcript in quantitative PCR tests to screen for colon cancer in outpatient settings. Moreover, given its higher expression in colon cancer tissues and the stability of cell-free RNA in circulation, the PHF19-207 transcript could potentially support the early detection of colon cancer, which is critical for more effective treatments. As a non-invasive RNA marker, the PHF19-207 transcript might offer a replacement for more invasive techniques, such as colonoscopy. However, additional research is necessary to confirm whether the expression levels of the PHF19-207 transcript in blood correlate with those in tumor tissues and whether its detection remains reliable in the presence of inflammation or other confounding factors.

## 5. Conclusions

Based on the findings of this study, the PHF19-207 transcript shows strong potential as a diagnostic biomarker for colon cancer, particularly for distinguishing malignant from non-malignant tissue. However, its role as a standalone biomarker for CRC screening is limited due to its potential overlap with inflammation-related changes in the mucosa. The results support further investigation into the development of non-invasive tests, possibly leveraging cell-free RNA detection, which could enhance early screening strategies. Additionally, combining analysis of the PHF19-207 transcript with other biomarkers, such as acute phase inflammation parameters, may improve its specificity and clinical applicability. Further research is essential to better understand the influence of inflammation on the PHF19-207 transcript expression and to refine its potential use in comprehensive CRC screening approaches.

## Figures and Tables

**Figure 1 biomolecules-15-00766-f001:**
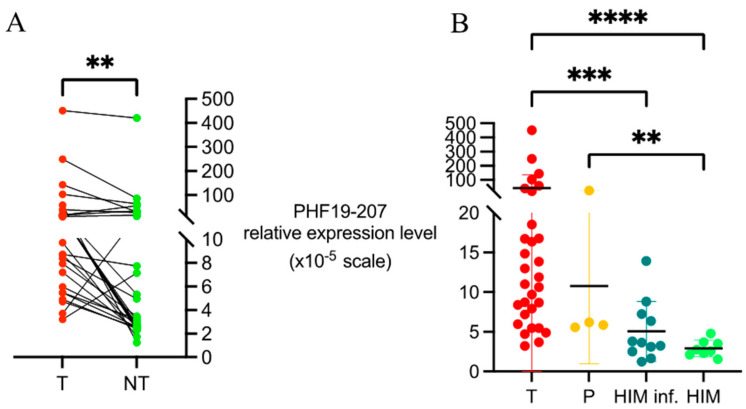
Relative expression of the PHF19-207 transcript in colon tissue samples. (**A**) Paired tumor and adjacent non-tumor tissue samples. (**B**) Tumor tissue samples, polyps, and samples of healthy mucosa with or without inflammation. Data are presented as 2^−dCt^ values and y-axes are scaled as 10^−5^. Horizontal lines represent mean ± standard deviation. ** *p* < 0.01; *** *p* < 0.001; **** *p* < 0.0001. T—tumor tissue; NT—adjacent non-tumor tissue; P—polyp; HIM inf.—healthy intestinal mucosa with inflammation; HIM—healthy intestinal mucosa without inflammation.

**Figure 2 biomolecules-15-00766-f002:**
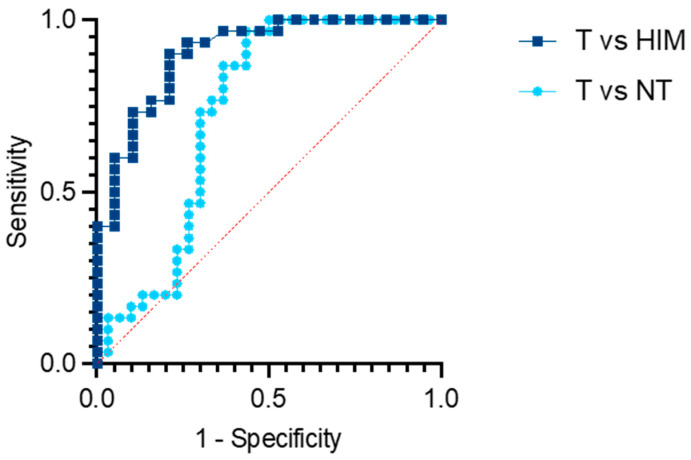
Receiver operating characteristic (ROC) curve for the prediction of colon cancer based on the expression levels of the PHF19-207 transcript in gastrointestinal mucosa. T—tumor tissue; NT—adjacent non-tumor tissue; HIM—healthy intestinal mucosa.

**Table 1 biomolecules-15-00766-t001:** Demographic and clinical data of the colon cancer patients.

Age (years), mean ± standard deviation	67.7± 7.2
Gender, n (%)	
Males	22 (73.3)
Females	8 (26.7)
Tumor localization, n (%)	
Right colon	18 (60.0)
Left colon	12 (40.0)
T stadium, n (%)	
T2	4 (13.4)
T3	19 (63.3)
T4	7 (23.3)
N stadium, n (%)	
N0	17 (56.7)
N1	7 (23.3)
N2	6 (20.0)
Lymphatic vessel invasion, n (%)	
Lx	1 (3.3)
L0	17 (56.7)
L1	12 (40.0)
Vein invasion, n (%)	
V0	17 (56.7)
V1	13 (43.3)
Perineural invasion, n (%)	
PN0	2 (6.7)
PN1	28 (93.3)
CEA(IU/mL), mean ± standard deviation	5.3 ± 4.7

## Data Availability

The data generated in this study are available from the corresponding author upon reasonable request.

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
