# Peer review of "Transcript PHF19-207 as a Potential Biomarker for Colon Cancer Diagnosis and Screening"

_biomolecules, 2025, doi:10.3390/biom15060766_

Round 1
Reviewer 1 Report
Comments and Suggestions for Authors
The article entitled “Transcript PHF19-207 as a potential biomarker for colon cancer diagnosis and screening” by Stefan Kmezic et al. is overall well written, introduced and analyzed; however, it does not provide strong evidence for the use of PHF19-207 mRNA as an screening method for CRC diagnosis.
- Incidence of CRC is high enough to take 2-3 surgical samples/week. So it is not justified the low sample size used along this research. Furthermore, authors have not shown sample size determination. Sample size is not enough to get this conclusion.
- Table of patient characteristics must show number of patients and percentages.
- Diagnostic potential of PHF19-207 must be compared to other FDA approved biomarker like CEA. Is better than CEA? Could the combination of PHF19-207 + CEA increase the accuracy of CEA alone?
- mRNA expression levels have no clinical utility. I strongly recommend validating this result at protein level by IHC analyses of resected samples since high expression protein levels of PHF19 is associated to survival in several types of tumors including CRC.
Author Response
Dear Sir,
We would like to valuable feedback, helpful comments, and constructive suggestions. Below is our point-by-point response to the remarks. All changes have been incorporated into the revised manuscript and are highlighted accordingly.
- Incidence of CRC is high enough to take 2-3 surgical samples/week. So it is not justified the low sample size used along this research. Furthermore, authors have not shown sample size determination. Sample size is not enough to get this conclusion.
Authors’ response:
We appreciate the reviewer’s thoughtful comment. We agree that larger studies are essential for the clinical translation of biomarkers. However, the sample size in this study was determined based on a targeted design to evaluate the PHF19-207 transcript expression in specific patient subgroups. Our aim was not to propose PHF19-207 as a definitive diagnostic or screening marker but to explore its potential relevance in an exploratory pilot study.
Although the sample size is limited, our results show statistically significant and biologically relevant differences, particularly the comparison between tumor samples and healthy intestinal mucosa, which demonstrated a high diagnostic performance (AUC = 0.9044, 95% CI = 0.8179–0.9908, p < 0.0001). The narrow confidence interval and strong statistical significance indicate that the sample size was sufficient to detect a meaningful effect size.
To address this comment, we clarified the focus of the study and emphasized the sample size limitation in the revised manuscript:
Introduction section
Lines 128-131 “In the current study, we focused on evaluating the clinical relevance of PHF19-207 transcript in patient-derived colon tissue samples. The aim of the study was to analyze the expression of PHF19-207 transcript in various types of colon tissue samples and to assess its potential as a biomarker for the early diagnosis of colon cancer. “
Discussion section
Lines 278-282 “This study was designed as a targeted, exploratory analysis focusing on expression patterns in two groups: colon cancer patients and individuals from the national CRC screening program. The primary objective was to evaluate PHF19-207 transcript as a potential biomarker for colon cancer. To explore its clinical relevance, the abundance of this transcript was analyzed in samples from two different settings.”
Lines 327-334 “While larger studies are undoubtedly essential for the clinical translation of biomarkers, the findings from this relatively small cohort provide compelling preliminary evidence supporting the potential of PHF19-207 transcript as a biomarker. Notably, its expression was consistently and significantly elevated in tumor tissue compared to matched non-tumor or healthy mucosa samples, demonstrating strong discriminatory power. The obtained AUC value of 0.9044 indicates the potential of PHF19-207 transcript to serve as a discriminative biomarker between malignant and healthy tissue, in line with general biomarker evaluation criteria.”
- Table of patient characteristics must show number of patients and percentages.
Authors’ response:
Taking the reviewer’s comment into consideration, we have revised Table 1 and included additional data. During this process, we identified several calculation errors, which we have corrected accordingly in the revised version.
Table 1. Demographic and clinical data of the colon cancer patients
|
Age (years), mean±standard deviation |
67.7± 7.2 |
|
Gender, n(%) |
|
|
Males |
22(73.3) |
|
Females |
8(26.7) |
|
Tumor localization, n(%) |
|
|
Right colon |
18(60.0) |
|
Left colon |
12(40.0) |
|
T stadium, n(%) |
|
|
T2 |
4(13.4) |
|
T3 |
19(63.3) |
|
T4 |
7(23.3) |
|
N stadium, n(%) |
|
|
N0 |
17(56.7) |
|
N1 |
7(23.3) |
|
N2 |
6(20.0) |
|
Lymphatic vessels invasion, n(%) |
|
|
Lx |
1(3.3) |
|
L0 |
17(56.7) |
|
L1 |
12(40.0) |
|
Vain invasion, n(%) |
|
|
V0 |
17(56.7) |
|
V1 |
13(43.3) |
|
Perineural invasion, n(%) |
|
|
PN0 |
2(6.7) |
|
PN1 |
28(93.3) |
|
CEA(IU/mL), mean±standard deviation |
5.3±4.7 |
- Diagnostic potential of PHF19-207 must be compared to other FDA approved biomarker like CEA. Is better than CEA? Could the combination of PHF19-207 + CEA increase the accuracy of CEA alone?
Authors’ response:
We thank the reviewer for this important point. In our study, PHF19-207 expression was assessed in paired tumor and adjacent non-tumor tissues, while CEA levels were measured in serum only in patients with confirmed colon cancer. As a result, serum CEA values were not available for individuals in the screening group, and CEA levels could not be directly matched to malignant versus non-malignant tissue expression, limiting the ability to perform a direct ROC comparison or to construct a combined diagnostic model (CEA + PHF19-207) across all study groups. As already stated in the manuscript, the outpatient clinic responsible for screening performs colonoscopy exclusively, without additional laboratory testing. Nevertheless, to partially address the suggestion, we performed a correlation analysis between serum CEA levels and PHF19-207 expression in tumor tissues of colon cancer patients. We included additional analysis and revised the manuscript:
Material and Methods section
Lines 200-202 “A non-parametric Spearman's rank correlation coefficient (rs) was applied to assess the correlation between variables.”
Results section
Lines 249-251 “No significant correlation was found between serum CEA levels and PHF19-207 transcript expression in tumor tissues (rs = -0.228, 95% CI = -0.5519 - 0.1544, p = 0.224).”
Discussion section
Lines 334-340 “The lack of correlation between serum CEA levels and PHF19-207 transcript expression in tumor tissues suggests that PHF19-207 may serve as an independent biomarker from CEA. However, the overlap of expression levels in inflamed healthy mucosa and premalignant polyps underscores a limitation in distinguishing these conditions, suggesting that PHF19-207 transcript alone may not suffice for early-stage detection and should be considered in combination with other clinical or molecular markers.”
Lines 342-344 “However, the combined diagnostic potential of clinical markers and PHF19-207 transcript should be further evaluated.”
mRNA expression levels have no clinical utility. I strongly recommend validating this result at protein level by IHC analyses of resected samples since high expression protein levels of PHF19 is associated to survival in several types of tumors including CRC.
Authors’ response:
We thank the reviewer for the suggestion to validate our findings at the protein level, and we agree that protein expression is often an important aspect of evaluating clinical relevance. However, we would like to emphasize that the PHF19-207 transcript analyzed in our study is not the primary protein-coding isoform of the PHF19 gene. According to the Ensembl database, PHF19-207 is annotated as a protein-coding transcript, but with an incomplete coding sequence (CDS), and it does not encode a full-length functional protein. Therefore, protein-based validation approaches, such as immunohistochemistry, are not applicable in this specific context.
While we recognize that mRNA-based biomarkers face challenges in clinical translation, growing evidence supports the potential role of non-coding RNAs, as useful diagnostic and prognostic tools in cancer. Given the characteristics of PHF19-207, it is more likely to function as a long non-coding RNA (lncRNA).
We believe that the significant expression differences observed in clinical tissue samples support the relevance of PHF19-207 transcript as a biomarker candidate. We have clarified the lack of functional protein associated with the PHF19-207 transcript and revised the text accordingly. We also included transcript structure scheme in Figure S1.
Introduction section
Lines 93-104 “According to the Ensembl database, the PHF19-207 transcript is annotated as a protein-coding isoform with an incomplete coding sequence (CDS) (the PHF19-207 transcript structure is provided in Figure S1). Previous in silico analysis has indicated a low coding potential for PHF19-207, suggesting that it may function primarily as a non-coding or partially coding RNA species with putative regulatory roles. Despite its protein-coding annotation, PHF19-207 encodes a truncated polypeptide lacking critical domains present in the full-length canonical PHF19 protein, thereby raising questions about its capacity to exert full protein function. This raises the possibility that PHF19-207 transcript may serve a regulatory function, potentially acting through RNA-mediated mechanisms such as competitive endogenous RNA interactions, modulation of RNA-binding protein activity, or the regulation of transcriptional and post-transcriptional processes. “
Lines 119-120 “The PHF19-207 transcript (888 bp in length), as a lncRNA, may serve as an example of this emerging trend.”
Reviewer 2 Report
Comments and Suggestions for Authors
The study is of reasonable design and the results are of clinical significance. There are several minor points related to the clarity of the manuscript:
- In the Ensembl database, PHF19-207 is annotated as a protein-coding transcript with an incomplete coding sequence (CDS). Did the authors classify this transcript as non-coding RNA based on its incomplete CDS? If so, a transcript structure scheme could be provided to show this. In addition, CPC2 and other tools can assess the protein-coding potential of a transcript. The authors would also evaluate PHF19-207 using these tools.
- A workflow scheme is needed to directly illustrate the sample grouping of this study. This is important for readers to smoothly recognize the clinical significance of this study
- The authors argued that PHF19-207 is not associated with tumor stage, but did not show detailed data behind this argument. Moreover, is PHF19-207 associated with patient prognosis based on the observation of the authors' cohort or other public datasets?
Author Response
Dear Sir,
We would like to valuable feedback, helpful comments, and constructive suggestions. Below is our point-by-point response to the remarks. All changes have been incorporated into the revised manuscript and are highlighted accordingly.
- In the Ensembl database, PHF19-207 is annotated as a protein-coding transcript with an incomplete coding sequence (CDS). Did the authors classify this transcript as non-coding RNA based on its incomplete CDS? If so, a transcript structure scheme could be provided to show this. In addition, CPC2 and other tools can assess the protein-coding potential of a transcript. The authors would also evaluate PHF19-207 using these tools.
Authors’ response:
Thank you for this insightful comment. Due to incomplete coding sequence (CDS), PHF19-207 transcript does not encode a full-length functional protein. We agree that clarification regarding the coding potential of PHF19-207 is important. Previously presented preliminary findings supported the hypothesis that PHF19-207 might act as a long non-coding RNA (lncRNA) by indicating its low coding potential. We have clarified the coding potential of PHF19-207 in the manuscript, and cited the relevant source. In addition, we have included a transcript structure scheme adopted from Ensembl database (Supplementary material: Figure S1) to illustrate the incomplete coding potential of PHF19-207. We have now revised the manuscript accordingly:
Introduction section
Lines 93-104 “According to the Ensembl database, the PHF19-207 transcript is annotated as a protein-coding isoform with an incomplete coding sequence (CDS) (the PHF19-207 transcript structure is provided in Figure S1). Previous in silico analysis has indicated a low coding potential for PHF19-207, suggesting that it may function primarily as a non-coding or partially coding RNA species with putative regulatory roles. Despite its protein-coding annotation, PHF19-207 encodes a truncated polypeptide lacking critical domains present in the full-length canonical PHF19 protein, thereby raising questions about its capacity to exert full protein function. This raises the possibility that PHF19-207 transcript may serve a regulatory function, potentially acting through RNA-mediated mechanisms such as competitive endogenous RNA interactions, modulation of RNA-binding protein activity, or the regulation of transcriptional and post-transcriptional processes. “
- A workflow scheme is needed to directly illustrate the sample grouping of this study. This is important for readers to smoothly recognize the clinical significance of this study
Authors’ response:
In accordance with the suggestion, we considered it most appropriate to include a graphical abstract, which we have now provided to enhance the clarity and visual summary of our work.
- The authors argued that PHF19-207 is not associated with tumor stage, but did not show detailed data behind this argument. Moreover, is PHF19-207 associated with patient prognosis based on the observation of the authors' cohort or other public datasets?
Authors’ response:
We clarified that we analyzed the association between the expression levels of PHF19-207 transcript and T and N stages in patients’ samples. We also reformulated the previously suggested association between PHF19-207 transcript expression and tumor stage. In our cohort, we were not able to assess the prognostic significance of PHF19-207 transcript expression, as clinical follow-up data, including patient survival outcomes, were not available. To avoid any potential misunderstanding, we have revised the paragraph that included term “prognosis”.
According to this comment, the manuscript is revised:
Results section
Lines 246-249 “No association was observed between the PHF19-207 transcript expression and tumor T or N stage ((T2: 2-dCt = 0.0014 ± 0.0021, T3: 2-dCt = 0.0003 ± 0.0005, T4: 2-dCt = 0.0001± 0.0001; p = 0.237) and (N0: 2-dCt = 0.0005 ± 0.0012, N1: 2-dCt = 0.0004 ± 0,0005, N2: 2-dCt = 0.0001 ± 0.0001; p = 0.497)).”
Discussion section
Lines 302-305 “Although preliminary in vitro data indicated higher expression levels in cell lines derived from advanced-stage colon tumors, our study did not find a significant association between PHF19-207 transcript expression and the T or N stage in patient samples.”
Lines 407-415 „In contrast, liquid biopsies that detect circulating tumor RNA offer a less invasive means of obtaining molecular data, potentially eliminating the need for surgical procedures. Detecting RNA molecules such as PHF19-207 in circulation may enable earlier disease recognition, potentially before symptoms emerge or imaging reveals pathological changes. Moreover, regular monitoring of circulating RNA levels might allow for tracking disease dynamics and adjusting therapeutic strategies accordingly. However, broader validation studies are required to establish the clinical utility of RNA biomarkers in the bloodstream, including PHF19-207 transcript.“
Reviewer 3 Report
Comments and Suggestions for Authors
Dear Authors
Manuscript describes well that PHF19-207 transcript shows strong potential as a diagnostic biomarker for colon cancer, particularly for distinguishing malignant from non-malignant tissues.
The first study group consisted of 30 surgical patients and 23 asymptomatic individuals (age 37 - 80 years, 56% males) who underwent national screening for CRC and were referred to colonoscopy due to positive fecal occult blood test.
The following steps should provide more clear information for readers to enjoy it
1) Study samples are very small size.
2) Please include CRC cell lines data – i.e., https://core.ac.uk/download/580032185.pdf
Human CRC tissue samples and CRC cell lines combine, it can give solid evidence to conclude your results.
Author Response
Dear Sir,
We would like to valuable feedback, helpful comments, and constructive suggestions. Below is our point-by-point response to the remarks. All changes have been incorporated into the revised manuscript and are highlighted accordingly.
- Study samples are very small size.
Authors’ response:
We appreciate the reviewer’s comment. Although the sample size is limited, study was designed to include well-matched tumor and non-tumor tissue pairs from the same patients and carefully selected healthy tissue from individuals undergoing colonoscopy screening, with RNA quality control across all samples. Our results demonstrated statistically significant and biologically relevant differences. According to this concern, we emphasized the sample limitation in the Discussion section.
Lines 327-334 “While larger studies are undoubtedly essential for the clinical translation of biomarkers, the findings from this relatively small cohort provide compelling preliminary evidence supporting the potential of PHF19-207 transcript as a biomarker. Notably, its expression was consistently and significantly elevated in tumor tissue compared to matched non-tumor or healthy mucosa samples, demonstrating strong discriminatory power. The obtained AUC value of 0.9044 indicates the potential of PHF19-207 transcript to serve as a discriminative biomarker between malignant and healthy tissue, in line with general biomarker evaluation criteria.”
- Please include CRC cell lines data – i.e., https://core.ac.uk/download/580032185.pdf
Authors’ response:
We thank the reviewer for this valuable suggestion. The results concerning colon cancer cell lines, which were previously presented, are part of a separate ongoing study focused on functional analyses in cell line models conducted by another researcher. As this manuscript focuses on the clinical relevance of the PHF19-207 transcript, these experimental data have not been included. Accordingly, we have emphasized the primary aim of the study in the revised manuscript. Nevertheless, in the revised manuscript, we have incorporated the observation that PHF19-207 transcript expression is upregulated in malignant colon cell lines compared to a non-malignant line, and potentially associated with tumor stage. We have cited the relevant source. To the best of our knowledge, no additional data regarding this transcript have been reported in the literature.
We revised the manuscript accordingly:
Introduction section
Lines 104-106“Furthermore, increased expression of PHF19-207 transcript was observed in malignant colon cancer cell lines compared to a non-malignant control, further supporting its potential role in colon cancer pathophysiology.”
Discussion section
Lines 302-305 “Although preliminary in vitro data indicated higher expression levels in cell lines derived from advanced-stage colon tumors, our study did not find a significant association between PHF19-207 transcript expression and the T or N stage in patient samples.”
Round 2
Reviewer 1 Report
Comments and Suggestions for Authors
After the first review round of the article entitled “Transcript PHF19-207 as a potential biomarker for colon cancer diagnosis and screening” by Stefan Kmezic et al., some amendments have been introduced; however, some others required further explanation.
- Authors said, “the sample size in this study was determined based on a targeted design to evaluate the PHF19-207 in specific patient subgroups”. We could admit that 23 asymptomatic individuals is the specific subgroup and could be difficult to recruit. However, 30 surgical patients are not representative of the CRC patient population. On the other hand, no justification of sample size has been included after this revision. If sample size is not justified no conclusion could be obtained from this study.
- In Statistical Analysis section author mention: “A non-parametric Spearman's rank correlation was applied…”; however, no normality test has been applied.
- Correlation with CEA expression was assessed but no statistical result was obtained. Interestingly, R2 = -0.228 seems to show a slight negative not significant correlation, that justifies the need to include more individuals in this study. Authors should include a limitation statement due to the limited sample size of their study.
- PHF19-207 could be acting as a long non-coding RNA (lncRNA). Could this finding be implemented in clinical practice? Authors should discuss this aspect in the discussion section not only in the introduction.
Author Response
We appreciate the reviewer’s comments. We have addressed each comment point-by-point and revised the manuscript accordingly to improve clarity, coherence, and overall quality. New changes are highlighted in green.
After the first review round of the article entitled “Transcript PHF19-207 as a potential biomarker for colon cancer diagnosis and screening” by Stefan Kmezic et al., some amendments have been introduced; however, some others required further explanation.
- Authors said, “the sample size in this study was determined based on a targeted design to evaluate the PHF19-207 in specific patient subgroups”. We could admit that 23 asymptomatic individuals is the specific subgroup and could be difficult to recruit. However, 30 surgical patients are not representative of the CRC patient population. On the other hand, no justification of sample size has been included after this revision. If sample size is not justified no conclusion could be obtained from this study.
Taking the comment into account, we clarified the inclusion period and patient selection criteria in the Materials and Methods, and further addressed these aspects and study limitations in the Discussion.
It is important to emphasize that the study was conducted at a single department of the Clinic for Digestive Surgery, a tertiary healthcare facility with a primary focus on rectal cancers and multivisceral colonic resections for locally advanced or metastatic disease. As such, patients with treatment-naive, non-disseminated colon cancer comprise approximately 25–30% of the total CRC cases treated annually at the department. This proportion aligns with the expected patient volume in this clinical context. The sample size was additionally impacted by some eligible patients choosing not to participate.
We revised the manuscript accordingly:
Materials and Methods
Lines 140-141 “This study included two groups of participants recruited at the Clinic for Digestive Surgery‐First Surgical Clinic, University Clinical Center of Serbia.“
Lines 145-153 “The first study group consisted of 30 patients (age 54–83 years, 73% males) who underwent surgery for colon cancer between April 2023 and February 2024. Inclusion criteria for patients were: histologically confirmed colon adenocarcinoma, absence of distant metastases, no preoperative treatment, and signed informed consent. Patients diagnosed with rectal cancer or those who declined to provide informed consent were excluded from the study. The second group included 23 asymptomatic individuals (age 37–80 years, 56% males) enrolled through the national CRC screening program conducted on an annual basis, with recruitment occurring between October 2022 and October 2023. These individuals were referred for colonoscopy following a positive fecal occult blood test.“
Discussion
Lines 288-299 “This study was designed as a targeted, exploratory analysis focusing on expression patterns in two groups: colon cancer patients and individuals from the national CRC screening program. The primary objective was to provide an initial assessment of the clinical applicability of PHF19-207 transcript, building on existing evidence that it is significantly overexpressed in malignant colon tissue compared to non-malignant tissue. To explore its clinical relevance, the abundance of this transcript was analyzed in samples from two different settings. Surgical samples of tumor and adjacent non-tumor tissue from non-metastatic, treatment-naive colon cancer patients were analyzed to evaluate the diagnostic potential of PHF19-207 transcript and its association with tumor characteristics. Patients with rectal cancer were intentionally excluded to minimize clinical heterogeneity and ensure a more uniform cohort in terms of tumor localization.“
Lines 306-309 “The relative abundance of PHF19-207 transcript was initially evaluated in matched tumor and adjacent mucosa samples collected during surgery. The inclusion of treatment-naive patients ensured that expression levels reflected intrinsic tumor biology, unaffected by prior therapy.”
Lines 342-349 “The main limitation of this study is its relatively small sample size, which resulted from the strict inclusion criteria. The study was specifically designed to evaluate a pre-identified biomarker as a potential screening tool, and therefore included only treatment-naive colon cancer patients without metastatic disease as they are representative of cases that would typically be identified through screening programs. This pilot study was conducted at a single department within a tertiary surgical center, where a large proportion of CRC cases involve either locally advanced rectal cancer treated with neoadjuvant therapy or metastatic disease, thus limiting the pool of eligible participants.”
- In Statistical Analysis section author mention: “A non-parametric Spearman's rank correlation was applied…”; however, no normality test has been applied.
In the revised version of the manuscript, we have updated the sentence in the Material and Methods section to explicitly state that the selection of statistical tests was based on the distribution of data:
Lines 202-204 “Statistical tests were selected based on the distribution of continuous data, which was assessed using the Shapiro–Wilk test.”
- Correlation with CEA expression was assessed but no statistical result was obtained. Interestingly, R2 = -0.228 seems to show a slight negative not significant correlation, that justifies the need to include more individuals in this study. Authors should include a limitation statement due to the limited sample size of their study.
In line with this comment, we included limitation of the study and reformulated the paragraph in the Discussion section:
Lines 342-363 “The main limitation of this study is its relatively small sample size, which resulted from the strict inclusion criteria. The study was specifically designed to evaluate a pre-identified biomarker as a potential screening tool, and therefore included only treatment-naive colon cancer patients without metastatic disease as they are representative of cases that would typically be identified through screening programs. This pilot study was conducted at a single department within a tertiary surgical center, where a large proportion of CRC cases involve either locally advanced rectal cancer treated with neoadjuvant therapy or metastatic disease, thus limiting the pool of eligible participants. Nevertheless, the consistent and statistically significant elevation of PHF19-207 transcript in tumor tissue compared to matched non-tumor or healthy mucosa samples suggests strong discriminatory potential. The obtained AUC value of 0.9044 supports the potential of PHF19-207 as a biomarker candidate and indicates that further validation may be beneficial. The lack of a statistically significant correlation between serum CEA levels and PHF19-207 transcript expression in tumor tissues suggests that PHF19-207 may function as a biomarker independent of CEA. However, this result should be re-evaluated in a larger cohort due to the observed weak inverse relationship. Furthermore, the overlapping expression levels seen in inflamed mucosa and premalignant polyps underscore a key limitation in distinguishing these conditions. This suggests that PHF19-207 transcript alone may not be sufficient for early-stage detection and should ideally be evaluated in conjunction with other clinical or molecular markers. In our study, such results were not available for consideration since the protocol in the outpatient clinic that manages individuals from the screening does not include any analyses other than colonoscopy.”
- PHF19-207 could be acting as a long non-coding RNA (lncRNA). Could this finding be implemented in clinical practice? Authors should discuss this aspect in the discussion section not only in the introduction.
In response to the comment regarding the interpretation of PHF19-207 as a potential lncRNA and its clinical relevance, we have reorganized and revised the relevant section of the Discussion. Specifically, we merged and reformulated the original paragraphs 5 to 8 to improve clarity and logical flow. This revised section has been relocated to follow the paragraph beginning with “While this study highlights the diagnostic potential of PHF19-207 transcript…” and ending with “…as part of a more comprehensive and affordable screening program” (previously paragraph 9).
The manuscript has been revised accordingly:
Lines 376-414 “This study suggests that the PHF19-207 transcript may serve as a marker of colon malignancy, regardless of whether it has a direct functional role. According to previous data, the increased expression of PHF19-207 transcript in tumors can be attributed to tumor cells, but the exact cellular origin of the observed differences remains unclear (16). Given its potential role as a lncRNA, the cellular and subcellular localization of PHF19-207 could indicate a possible regulatory function. Since PHF19 contributes to chromatin remodeling through its PHD finger domain, the PHF19-207 as non-coding variant, may similarly function in the nucleus by regulating gene expression via interactions with chromatin or RNA-binding proteins. Conversely, if located in the cytoplasm, it might function through post-transcriptional mechanisms, such as modulating RNA stability or acting as a miRNAs sponge to regulate oncogene expression. The PHF19-207 transcript may be involved in tumor initiation or maintenance by influencing key signaling pathways associated with cell proliferation, apoptosis resistance, or immune evasion. These diverse potential roles emphasize the need for further research to identify its downstream targets and clarify the underlying molecular mechanisms. Importantly, future research should also investigate whether PHF19-207 expression is associated with tumor progression, which could provide deeper insight into its biological significance and reveal its potential as a prognostic biomarker or therapeutic target in colon cancer. Understanding whether the PHF19-207 transcript exerts a regulatory function in cancer-related cellular processes could shed light on its biological role and reveal whether its modulation may have therapeutic potential.
Moreover, the PHF19-207 transcript might be also suggested to participate in modulating the inflammatory response in the tumor microenvironment. Its elevated expression in inflamed compared to non-inflamed mucosa suggests that local inflammation may influence its expression. The link between inflammation and cancer is well-established, with chronic inflammation often driving tumorigenesis and influencing tumor progression. PHF19-207 transcript may be involved in this process by regulating immune cell recruitment, cytokine production, or other inflammatory mediators. If confirmed, this link would imply clinical relevance for PHF19-207 as a marker related to inflammation-associated premalignant lesions and as a potential target for modulating inflammation to reduce tumor progression. The interaction between tumor and stroma, along with the contribution of immune cells to the cancer microenvironment, could further influence its expression, making PHF19-207 a complex marker shaped not only by tumor cells but also by the surrounding tissue environment. This concept requires further investigation, particularly to clarify the extent to which PHF19-207 transcript expression is influenced by local inflammation or other non-tumor cells present in the colon tissue. Taken together, these observations support the potential of PHF19-207 as a biomarker candidate, while its clinical utility as a therapeutic target remains to be established.”
Reviewer 3 Report
Comments and Suggestions for Authors
Dear Authors, Revised manuscript are explains well.
Author Response
We appreciate the reviewer’s positive comment.